# Effective Strategies for Managing COVID-19 Emergency Restrictions for Adults with Severe ASD in a Daycare Center in Italy

**DOI:** 10.3390/brainsci10070436

**Published:** 2020-07-09

**Authors:** Natascia Brondino, Stefano Damiani, Pierluigi Politi

**Affiliations:** Department of Brain and Behavioral Sciences, University of Pavia, via Bassi 21, 27100 Pavia, Italy; stefano.damiani01@ateneopv.it (S.D.); pierluigi.politi@unipv.it (P.P.)

**Keywords:** autism spectrum disorder, COVID-19, challenging behavior

## Abstract

The COVID-19 pandemic has posed a serious challenge for the life and mental health of people with autism spectrum disorder (ASD). COVID-19 sanitary restrictions led to significant changes in the lives of people with ASD, including their routines; similarly, these modifications affected the daily activities of the daycare centers which they attended. The present retrospective study evaluated the impact of COVID-19 restrictions on challenging behaviors in a cohort of people with severe ASD attending a daycare center in Italy at the beginning of the pandemic. During the first two weeks of the pandemic, we did not observe variations in challenging behaviors. This suggests that adaptations used to support these individuals with ASD in adapting to the COVID-19 emergency restrictions were effective for managing their behavior.

## 1. Introduction

Autism spectrum disorders (ASDs) are complex neurodevelopmental conditions characterized not only by impairment in socio-emotional reciprocity and communication, but also by restrictive and repetitive patterns of behaviors (RRB) and interests [1]. Until recently, the RRB domain has received little attention compared to the social domain. Several mechanisms may underlie RRB, as they can act as a method of communicating needs or obtaining attention. In addition, RRB may often represent a self-regulatory system to modulate anxiety and the level of activation. While the exact mechanism underlying specific types of RRB may be relevant to guide treatment, the impact of RRB, irrespective of the etiopathogenesis, is particularly relevant in severe ASD with comorbid cognitive impairment: Individuals in this group often present an unpostponable adhesion to routines, rituals, and repetitive behaviors which could significantly impair the quality of life of both ASD subjects and caregivers [2]. Additionally, disruption of routines or, more generally, changes to expectations are usually correlated with a worsening in challenging behaviors [2,3] leading to bouts of self- and other-directed aggressiveness.

The emergence of the COVID-19 pandemic in Italy started in February and is still ongoing. At the beginning of the emergency, the Italian government promulgated several restrictions of increasing severity, starting on 23 February 2020, in order to flatten the curve of contagions. Firstly, the government closed schools, restaurants, malls (on the weekends), swimming pools, and gyms, forbade gatherings (no more than 8 people per room), and recommended social distancing of at least 1 m between each other. It was also recommended to frequently wash hands in order to contain the virus.

It is evident that the COVID-19 emergency restrictions resulted in many relevant changes in the lives of people with ASD and their families. Such a complex situation was extremely difficult to share with service users, especially when learning disabilities were present. Understanding why things were changing, why the sanitary limitations should be scrupulously followed, without losing hope that the emergency would end if everyone cooperates, has been a real challenge for people with developmental disabilities. Additionally, the recommended use of personal protective equipment (PPE) has been controversial among this population. In this paper we focus on a peculiar issue, the impact of COVID-19 restrictions on the routine of daycare centers for adults with ASD and cognitive impairment, which rely on maintaining predictability and an ecologically structured context. The application of governmental restrictions determined significant changes in the activities of the daycare center and a great effort from the healthcare workers to maintain the quality and specificity of services provided to people with ASD. In the end, the difficulties in applying COVID-19 emergency restrictions led to the closure of all daycare centers for people with developmental disabilities one week before total lockdown of the country, which happened on 10 March 2020. These changes may have had the potential to increase distress in our patients and, consequently, challenging behaviors and the need of psychotropic medications [4]. The countermeasures and modifications of daycare center activities were put in place in order to comply to COVID-19 restrictions. In spite of having been dictated by necessity, these restrictions were implemented and inspired by knowledge on management of RRB and problem behaviors. For instance, differential reinforcement of variability was put in place in order to reinforce changes in routine activities or in behavioral responding to ordinary situations. As the spontaneous rate of variability is extremely low in autism, therapists firstly reinforced non-repetitions, moving subsequently to reinforcing of behaviors that were not previously observed.

The aim of the present study is to evaluate the impact of restrictions on challenging behaviors in a sample of individuals with ASD and cognitive impairment attending a daycare center before complete lockdown.

## 2. Materials and Methods

We conducted a retrospective study evaluating medical charts in a daycare center for adolescents and adults with ASD.

The daycare center “Il Tiglio” is a day center specifically designed for individuals with severe ASD and comorbid cognitive impairment, located in the Lombardy Region, Italy, near a regional nature park. It accommodates 18 individuals with ASD, and the staff is composed of 7 therapists, 1 psychologist, 1 kinesiologist, 2 care assistants, 1 nurse, and 1 consultant psychiatrist. Weekly schedules for each individual are strictly monitored and kept consistent. Planned activities consist of physical activity (water-based activities in a nearby swimming pool once a week, trekking once a week, judo lessons once a week, adaptive physical activity twice a week—both at the daycare center and in a nearby gym), horticultural therapy (once a week), cognitive training for language production and augmentative and alternative communication (AAC) use (every day), occupational therapy (every day), and art therapy (twice a week). The center is open five days a week from 8:00 a.m. to 5:00 p.m.

To comply with COVID-19 emergency restrictions, several adjustments were adopted in order to maintain the best quality of life for individuals with ASD. Specifically, as swimming and physical contact sports were no longer possible to perform, we implemented trekking (which was done every day for at least 1–2 h). Laboratories were split so that no more than 4–5 people were in the same room at the same time. AAC and social stories were implemented in order to provide information about the virus and about specific restrictions (for instance, some of them usually went to malls on the weekends with their parents as part of their routine and, as shops were closed, this could have resulted in potential problem behaviors). All individuals with ASD had already learned how to wash their hands thanks to a behavioral treatment and positive reinforcement; therefore, this routine was increased in frequency and inserted in a visual agenda for each of them. To increase changes in routine behaviors, differential reinforcement of variability was implemented: During the time between the arrival at the day center and the beginning of activities, everyone had a specific routine which could include doing puzzles, drawing, or running in the field outside. Every time the individual with ASD changed his/her arrival routine, a reinforcement (consisting in the possibility to listen to music or watching preferred parts of movies or social reinforcement) was given.

### 2.1. Measures

As general clinical practice, each day care workers filled out the aberrant behavior checklist (ABC) [5] for each individual with ASD. The ABC is a scale empirically designed to measure psychiatric symptoms and problem behaviors in subjects with developmental disabilities. It is composed of 58 items, evaluating 5 domains: Irritability; lethargy/social withdrawal; stereotypic behavior; hyperactivity; and inappropriate speech. Higher scores indicate a higher level of problem behaviors. The ABC is generally completed by the primary caregiver and informants can complete the scale in 10–15 min the first time they fill it in; subsequently it is more rapid, and it is used in our daycare center on a daily basis. The ABC was specifically designed for subjects living in institutions and residential settings, and therefore appeared appropriate for our center.

### 2.2. Design

We retrospectively evaluated the effect of COVID-19 emergency restrictions and daycare center implementation on problem behaviors using our registry. We evaluated changes in ABC total scores between 19 February 2020 (the last day free from restrictions) and 4 March 2020 (two weeks after the full restrictions were applied).

### 2.3. Statistical Analysis

Differences between the two time points were evaluated by means of a paired-sample *t*-test after the normality of data was ascertained. To account for multiple comparison, Bonferroni’s correction was applied. IBM SPSS Statistics for Windows, version 23 (IBM Corp., Armonk, NY, USA) was used for all statistical analyses. Two-tailed *p*-value < 0.06 was regarded as statistically significant.

## 3. Results

General characteristics of the sample are reported in Table 1. Our sample was composed of 18 young adults, of which 13 are males. All presented severe ASD; four individuals with ASD occasionally showed self-injurious behavior, while seven had severe bouts of aggression. All individuals were diagnosed during childhood. However, each of them was also re-evaluated by a senior psychiatrist with specific expertise in ASD in adulthood, according to the DSM 5 criteria. All individuals were rated as Level 3 of severity (requiring very substantial support). Diagnosis was supported by ADI-R [6] in all cases. ADOS 2 module 4 [7] was also performed in a verbal subject with mild cognitive impairment (*n* = 1). Cognitive impairment was evaluated by means of the Leiter 3 scale and by clinical judgment. Most of the sample subjects received stable psychiatric medications. Antipsychotics were administered to more than half of the sample (aripiprazole *n* = 4, risperidone *n* = 1, olanzapine *n* = 1, levomepromazine *n* = 2, clotiapine *n* = 4), while mood stabilizers were less frequently prescribed (valproate *n* = 3, gabapentin *n* = 2).

Overall, no significant differences in ABC scores were observed between the two time-points (24.83 ± 11.75 vs. 22.33 ± 12.18, t = −1.07, *p* = 0.29). In order to evaluate the generalizability of our findings, our baseline data (t0: 19 February 2020) were compared to the same period of the previous two years (19 February 2019 and 19 February 2018). No significant differences were found (2019 t = −0.09, *p* = 0.43; 2018 t = 0.03, *p* = 0.46). Additionally, we compared the 19 February 2020 ABC scores with ABC scores of the previous five days to determine if our baseline data were consistent with the previous week: No significant differences were found (*p* not significant for all the comparisons). We evaluated differences in ABC subscales: No statistically significant differences were found (see Appendix A).

## 4. Discussion

Our retrospective study did not show a significant change in problem behaviors in our individuals with ASD after COVID-19 restrictions were initiated. We could cautiously hypothesize that the preventive countermeasures we adopted were effective in reducing distress in individuals with ASD. This could be due to several reasons: First, maintaining the same amount of physical activity (increasing trekking over water-based activity or group contact sports) could have played a key role in controlling problem behaviors in individuals with ASD: It is well-known that physical activity can reduce aggressiveness, stereotyped and self-injurious behaviors, as well as self-stimulation [8]. Walking/trekking could be feasible for almost everyone, irrespective of physical fitness. It is cost-effective and it could be performed outdoors (both in winter and in summer), minimizing the risk of contagion connected with closed indoor environments, while enhancing health benefits connected with exposure to nature. Second, splitting the initial laboratory group into smaller groups (4 individuals or less) could have helped in maintaining the daily schedule without major deviations, allowing for social distancing at the same time. Third, smaller groups increased the ratio between therapists and individuals with ASD, leading to more personalized interventions. Fourth, reducing transfers to other facilities (like swimming pools, gyms) may have led to a lower sensory stimulation and therefore lower anxiety. Finally, AAC as well as social stories could have been a powerful instrument for changing the visual agenda of each individual and teaching new routines (i.e., wearing a facial mask) or enhancing already learned skills (i.e., hand washing).

Our retrospective study presents several limitations: The sample size is small, even if compatible with the average sample size of a daycare center. Another limitation is the lack of a control group. Unfortunately, the absence of a control group is inevitable, as sanitary restrictions were mandatory for all centers in the country. In the future we could design retrospective studies in order to evaluate time fluctuation in problem behaviors before and after the pandemic of COVID-19. In fact, the impact of COVID-19 after the lockdown and closure of daycare centers is yet to be determined. During lockdown, online interventions were implemented as suggested in [9] in order to help parents in caring for their loved ones, who did not have access to their daycare center. Additionally, the need for emergency home interventions, as well as dose-increases of psychiatric medications, were required in several cases. Another potential limitation regarded the care workers who filled the ABC and performed the intervention: Our implementation was designed by the coordinator of the daycare center who did not care directly for individuals with ASD and did not complete the ABC scores. All care workers were blind to the subsequent evaluation leading to the present study. Finally, the time interval chosen for the evaluation of the impact of COVID-19 restriction could have been longer in order to detect changes in behaviors; however, the ABC is designed to detect changes in a shorter period of time and the worsening of problem behaviors usually appear more rapidly than improvement.

At present, the COVID-19 emergency is still representing a massive psychological overload for all individuals, among whom we should not forget individuals with ASD, with their peculiar needs. Despite its limitations, our study may provide some important suggestions for the re-opening phase and the return to a “new normal”: Daycare centers in Italy are still struggling to find a compromise between minimizing the risk of contagion and providing adequate care for individuals with severe ASD. In fact, the long-term lack of specific interventions in this group may have resulted in loss of skills, reduction of positive behaviors, and increases in maladaptive behaviors. Additionally, returning to the same level of activity as before the COVID-19 pandemic may not be easy. For instance, changes due to restrictions and the general disempowerment of the public care services may slow the return to normality for individuals with ASD. On the other hand, the slow progress may be supportive for individuals with ASD, allowing them to re-adapt to a full daycare schedule at a slower pace.

## Figures and Tables

**Table 1 brainsci-10-00436-t001:** General characteristics of the sample.

Variables	Mean ± SD or % (*n*)
Age	22.72 ± 4.75
Gender, male	72.2% (13)
DSM 5 level of severity	
Level 3	100% (18)
Verbal behavior	
Verbal	44.4% (8)
Minimally verbal	27.8% (5)
Non-verbal	27.8% (5)
Cognitive impairment	
Mild	5.6% (1)
Moderate	50% (9)
Severe	22.2% (4)
Profound	22.2% (4)
Medications	83.3% (15)
Antipsychotics	66.6% (12)
Mood stabilizers	27.8% (5)

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
