# Peer review of "Effective Strategies for Managing COVID-19 Emergency Restrictions for Adults with Severe ASD in a Daycare Center in Italy"

_brainsci, 2020, doi:10.3390/brainsci10070436_

Round 1
Reviewer 1 Report
Thank you for this timely article in what has been a novel experience for everyone. As you point out, the needs of individuals with ASD are important to consider within the context of COVID-19 and should not be overlooked. Your work here is certainly timely and provides the field some preliminary information for what types of strategies and interventions might be helpful to assist individuals with ASD during this odd time. Below you will read more specific feedback about your submission.
Introduction-Your introduction stated very clearly the issues at hand. One suggestion to increase clarity for the reader is to explain more in-depth (towards the end of your last paragraph) by providing an example of the differential reinforcement techniques you discuss being applied. Additionally, I questioned whether that information might be more appropriate in the Materials and Methods section where are you outline other specific strategies used (visual schedules, use of Trekking etc). Although there may be some nuance between the two that I am missing or is lost in translation so to speak.
Materials and Methods- You clearly outlined the setting, subject characteristics as well as interventions utilized.
- One suggestion under the Measures subsection is to explicitly described how to interpret that ABC. For example, "higher scores indicate higher levels of problem behaviors". This will assist the reader in being able to more accurately interpret the results of your analyses. Additionally, it would be helpful to know if the care workers who completed the ABC scale were also those implementing the interventions with the individuals with ASD or not. As this could possibly introduce some bias responding.
- Under the Design subsection, I am left wondering why you decided to pick your second time point. So it would help the reader to understand more explicitly your thought process there. Some readers might argue that two weeks may not be enough time to see a change in behavior. Another thought was whether in addition to the overall Total ABC score, you noticed any differences across subscale scores (Irritability, Withdrawal, Social/Lethargy, Stereotypic Behavior, Hyperactivity, Inappropriate Speech) between the two time points.
Results- The characteristics of your sample and diagnostic characterization were clearly outlined and provide support to your argument of looking at individuals with ASD that would require more intensive community supports and with those who may have more difficulty with understanding the changes in routines COVID-19 has required. Your discussion of the interpretation of the statistical analyses were clear and I appreciated your providing additional data to evaluate the validity of your baseline data.
Discussion- The discussion nicely laid out your cautious interpretation of the results of the study. The limitations section may be expanded to include some of the potential issues described above (time points, biases in completing forms) where appropriate.
Reviewer 2 Report
The authors performed a retrospective study of the behaviors of 18 individuals with autism spectrum disorder (ASD) in a daycare facility for individuals with severe ASD before and two weeks after the initiation of severe restrictions by the government to minimize the spread of an infectious disease. They compared and contrasted scores on the Aberrant Behavior Checklist (ABC) before and two weeks after the restrictions. After identifying the normality of the distributions, the authors applied a two-sample t test to determine no significant differences in behaviors. They conclude that their substitution of alternative activities for the participants was an effective intervention to prevent the development of serious maladaptive behaviors due to the drastic changes in routine required by the governmental restrictions.
Overall the authors identify an important problem, the maintenance of a functional day program for severely impaired individuals with ASD and intellectual disability (ID) despite severe governmental restrictions required to minimize the spread of an infectious disease. The substitution of alternative activities by staff to follow the restrictions without behavioral deterioration is noteworthy. Readers likely will benefit from following the example of the authors. Likely the manuscript may merit publication with revisions.
The diagnostic data for the participants merits a full description. A table with the results of the ADI-R and ADOS components and the intellectual testing of each participant would be valuable, possibly as a supplementary file. Since some readers may be unfamiliar, please include references for the ADI-R and had the ADOS. Additionally specific data about the items of the ABC for each participant at the before and after time points would be valuable data as a supplementary file. Was any change noted in the sub scales of the ABC?
Sansone and colleagues (J Autism Dev Disord (2012) 42:1377–1392 DOI 10.1007/s10803-011-1370-2) developed a new set of sub scales of the ABC for people with fragile X syndrome (FXS). Since the author's population exhibits traits similar to those with FXS, application of Sansone's sub scales the study population may distinguish differences not yet detected.
While the current study population is small, the publication of the details of the data would provide readers the basis to compare and contrast findings with populations around the world.
Reference 4. Please list all authors.
This study is important for the world as a whole. There may be a second wave of this infectious disease in coming months. There may be novel infectious diseases in the future that will require comparable governmental restrictions. Readers will benefit from knowledge of effective techniques to prevent severe behavior problems in day programs for people with ASD and ID when governmental restrictions occur.
Reviewer 3 Report
This paper examined behavior changes after Covid-19 restrictions were imposed on a day care center serving adults with severe ASD in Italy. It is generally well-organized and is an interesting examination of a potentially problematic situation, but has many writing concerns. Its value and readability would be improved by addressing the following concerns:
- The title does not seem to accurately communicate the important factors of the study. A suggested alternate title is “Effective Strategies for Managing Covid-19 Emergency Restrictions for Adults with Severe ASD in a Day Care Center in Italy.”
- Throughout the paper, the words, “patients” or “subjects” or “study group” do not seem to be the best terms to identify these research participants. “Clients” or “research participants” are suggested. Terminology should be consistent each time.
- Many problems with colloquial or confusing language. See below.
- Abstract, line 14: need to define “lockdown.”
- Abstract, line 14: change to “This suggests that adaptations used to support these clients in adapting to the Covid-19 emergency restrictions were effective for managing their behavior.”\“Firstly,” “secondly,” “thirdly,” “lastly” are rather colloquial. Better to say “first,” “second,” “third,” “last or finally.”Page 2, line 52: change “lead” to “led.”
- Page 2, line 72: change “composed by” to “composed of.”
- Page 2, line 73: change “maintained constant” to “kept consistent.”
- Page 2, line 73: “Proposed activities” does not seem to be the correct terminology. Did you mean “Typical activities?”
- Page 2, line 73: change to “consist of physical…
- Page 2 & 3, Materials and Methods and Results: need to use past tense.
- Page 2, line 83: change to “social stories were implemented to provide…”
- Page 2, line 84: change to “went to malls on the weekends with…”
- Page 2, line 86: sentence starting with “Hand washing” is awkward. Needs to be rewritten.
- Page 2, line 93: change “filled” to “completed.”
- Page 3, line 94: change to “more rapid and it is used…”
- Page 3, line 96: change to “therefore, appeared appropriate for our center.”
- Page 3, line 106: change “calculation” to “statistical analyses.”
- Page 3, line 106: need to define the p-value for this study.
- Page 3, line 109: change “composed by” to “composed of.”
- Page 3, line 111: add a comma after “behavior.”
- Page 3, table 1: Level 3 needs to be defined.
- Page 4, Line 128: change to “previous five days to determine if our baseline data were consistent with the previous week…”
- Page 4, line 133: change to “restrictions were initiated.”
- Page 4, line 136: change “play” to “played.”
- Page 4, line 142: change to “laboratory group into smaller groups…”
- Page 4, line 147: add a comma after “stories.”
- Page 4, line 157: change “could” to “did.”
- Page 4, line 161: “forgive” is not the correct word. Please correct.
- Page 4, line 165: change to “specific interventions for this group may have resulted in loss of skills, reduction in positive behaviors…”
- Page 4, line 169: change to “normality for these individuals. On the other hand, the slow progress may be supportive for individuals with ASD, allowing them to…”
